# Energy Density and Level of Processing of Packaged Food and Beverages Intended for Consumption by Australian Children

**DOI:** 10.3390/nu17142293

**Published:** 2025-07-11

**Authors:** Sally MacLean, Kristy A. Bolton, Sarah Dickie, Julie Woods, Kathleen E. Lacy

**Affiliations:** 1School of Exercise and Nutrition Sciences, Deakin University, Geelong, VIC 3220, Australia; sallylmaclean@gmail.com; 2Institute for Physical Activity and Nutrition, Deakin University, Geelong, VIC 3220, Australia; kristy.bolton@deakin.edu.au; 3School of Clinical Sciences, Department of Nutrition, Dietetics and Food, Monash University, Melbourne, VIC 3168, Australia; sarah.dickie@monash.edu; 4Institute for Physical Activity and Nutrition, School of Exercise and Nutrition Sciences, Deakin University, Burwood, VIC 3125, Australia; j.woods@deakin.edu.au

**Keywords:** energy density, processing, food

## Abstract

**Background/Objectives**: Higher energy density (ED; kJ/g) and higher levels of processing of foods and beverages have been associated with childhood obesity and reduced diet quality. This study described and examined the distribution of ED and levels of processing of new food and beverage products intended for Australian children (0–4 years, 5–12 years). **Methods**: This study used 2013–2023 data from the Mintel Global New Products Database. Products were classified by ED (low ≦ 4.184 kJ/g, medium > 4.184 kJ/g and <12.552 kJ/g, or high ≧ 12.552 kJ/g) and level of processing (using the NOVA classification system; unprocessed/minimally processed foods; processed culinary ingredients; processed foods; ultra-processed foods (UPFs)). Non-parametric statistics were used to examine ED and level of processing by age and ‘Food’ and ‘Drink’ groups. **Results**: Of the 1770 products analysed, 56% were classified as high-ED and 81% as UPF. Among ‘Food’ products intended for children ‘5–12 years’, 93% were classified as UPFs. The differences in ED classification between non-UPFs and UPFs were significant for ‘Food’ products intended for children aged ‘0–4 years’ (*p* < 0.001) but not for children aged ‘5–12 years’ (*p* = 0.149). **Conclusions**: The prevalence of high-ED and UPFs in the Australian packaged food supply demonstrates the need to tighten regulations around products intended for children. The regulation of low-ED UPFs (i.e., recognised by Australian Dietary Guidelines as ‘healthy’) may also be required.

## 1. Introduction

Childhood obesity is a major public health concern [1]. In Australia, one quarter of children are living with overweight (17%) or obesity (8.1%) [2], with this prevalence having almost tripled since 1985 [3]. Childhood and adolescent obesity can significantly contribute to the development of a variety of other chronic conditions such as type 2 diabetes and cardiovascular disease [4,5]. A key modifiable risk factor of obesity is diet [6], with children aged 13 years and younger averaging over one third of total energy consumed per day from discretionary foods (i.e., ultra-processed foods, or foods high in sugar, saturated fats and/or energy density (ED)) [6,7]. ED and level of processing are widely used indicators of food and beverage nutritional quality [8,9,10,11] and are both associated with weight gain [12,13].

The energy density of a food or beverage is defined as the amount of energy per specific weight or volume of food or beverage, e.g., kJ/g or kJ/mL [14,15,16]. Fat content and water content are the strongest influencers of ED, providing ~37 kJ/g and 0 kJ/g, respectively [15,17], whilst carbohydrates and protein both provide ~17 kJ/g, and fibre provides ~8 kJ/g because it is not fully digested and absorbed [15,17]. ED is often a contrasting concept to nutrient density (nutrient per weight (g)) [18].

Multiple experimental studies have demonstrated a sustained and direct effect of ED on energy intake (EI) among children when the ED of a food or meal was modified [19,20,21,22,23]. Manipulating dietary energy density (DED) can be achieved by incorporating more fruits and vegetables into meals and/or reducing the fat content of foods, which results in reduced DED [20,22]. Regular overconsumption of high-ED foods may play an important role in the mediation of energy imbalance given young children living with obesity tend to consume a higher DED than children who are lean [12]. Recommended strategies in managing energy balance in children includes increased consumption of low-ED foods that are often low in fat and added sugars and high in fibre (i.e., fruits, vegetables, and low-fat dairy) and limited consumption of high-ED (e.g., non-core, discretionary) foods [6,8,9,19,20,24,25].

Ultra-processed foods (UPFs) are defined as ‘formulations of ingredients, mostly of exclusive industrial use, that results from a series of industrial processes’ [26]. Ultra-processed foods are the highest level of processing in the NOVA classification system, which categorises foods into four groups based on increasing levels of processing [26,27]. The level of processing refers to the mechanical process(es) undertaken to alter a food (e.g., grinding, drying, fermenting, etc.), the addition of industrial ingredients (e.g., cosmetic additives or processed substances) [28], and/or the addition of sugar or fat replacements [29], all of which enhance the texture, appearance, and/or taste of the food, whilst also extending product shelf-life [29]. Often UPFs have an extensively altered food matrix, making them nutritionally inadequate [27,30]. These processes can enhance the sensory characteristics of foods and beverages [31] by increasing their ED and palatability, which disrupt signalling pathways related to food control and appetite regulation [32] and may lead to overconsumption of these foods [28], increased EI, and childhood obesity [28,33].

The consumer food environment plays an important role in influencing the diet of children and provides insight into the quality of products that children are exposed to within their community [34]. This study will be the first to describe foods and beverages targeting children in Australia using both ED and NOVA, as well as whether foods of different levels of processing vary by energy density, over a 10-year period starting from the implementation of the 2013 Australian Dietary Guidelines (ADGs). Whilst the ADGs do not explicitly refer to ED and NOVA, they are both strong indicators of food quality. Analysing both ED and NOVA will provide a more comprehensive understanding of their effects on the quality of products. The results of this study can potentially be used to inform policy change to tighten regulations around the marketing of food and beverages intended for child consumption in Australia.

The aims of this study were to

Describe the ED and level of processing of packaged food and beverages intended for consumption by children (0–4 years and 5–12 years) introduced into the Australian food supply in the last 10 years (2013–2023).Examine the distribution of the ED classification by the level of processing of packaged food and beverages intended for consumption by children (0–4 years and 5–12 years) introduced into the Australian food supply in the past 10 years (2013–2023).

## 2. Materials and Methods

An observational descriptive analysis of packaged foods and beverages intended for children (0–4 years and 5–12 years), released into the Australian retail marketplace between May 2013 and May 2023, was conducted using the Mintel Global New Products Database (GNPD) [35]. Both ED and level of processing (using NOVA) were determined and analysed.

### 2.1. Mintel Global New Products Database

Mintel, a market research company, provides global coverage of newly released packaged products. The online Mintel GNPD provides updated product data from Mintel shoppers who access multiple data sources from over 80 countries [35]. Mintel GNPD provides information from major supermarket chains, and some independent chains, across Australia and reports on a range of product categories, including ‘Food’, ‘Drink’, ‘Pet’, ‘Health and Hygiene’, and ‘Specialised Nutrition’ [35]. Food and beverages are categorised into ‘Food’ or ‘Drink’ groups, with 17 and 8 subcategories, respectively. ‘Food’ subcategories include Baby Food, Bakery, Breakfast Cereals, Chocolate Confectionary, Dairy, Desserts and Ice Cream, Fruit and Vegetables, Meals and Meal Centres, Processed Fish, Meat and Egg Products, Sauces and Seasonings, Savoury Spreads, Side Dishes, Snacks, Soup, Sugar and Gum Confectionary, Sweet Spreads, and Sweeteners and Sugar. ‘Drink’ subcategories include Alcoholic Beverages, Carbonated Soft Drinks, Hot Beverages, Juice Drinks, Nutritional Drinks and Other, Ready to Drinks, Sports and Energy Drinks, and Water [35]. Products are allocated to a targeted demographic determined by Mintel (i.e., ‘0–4 years’ or ‘5–12 years’), which is based on packaging attributes with age-related positioning claims (e.g., nutrient and/or health claims, children’s graphics, or specific age guidance on the front of packaging) [35].

### 2.2. Data Collection and Cleaning

Mintel GNPD data on all newly released products in Australia between May 2013 and May 2023 were collected for the age demographics ‘0–4 years’ and ‘5–12 years’. ‘Food’ and ‘Drink’ categories were included in the data search, whilst ‘Pet’, ‘Health and Hygiene’, and ‘Specialised Nutrition’ (i.e., products formulated for targeted consumers with medical nutritional needs, including infant formula due to its specialised nature) were excluded. All ‘Food’ and ‘Drink’ subcategories were included in the data search except Alcoholic Beverages due to the target demographics of this research.

Product information extracted into Microsoft Excel included record ID, product, product variant, brand, market, category, subcategory, date published, product description, nutrition information, on-pack ingredients, energy (kJ/100 g or 100 mL), and demographic information (i.e., children ‘0–4 years’ and ‘5–12 years’). ‘Nutrition Information’ and ‘Energy (kJ/100 g or 100 mL)’ were extracted to calculate the ED. ‘On-pack ingredients’ were extracted to determine the NOVA classification of each product. Data were exported for data cleaning and analysis. Data were cleaned by removing duplicate items and anomalies. Where duplicates were products that contained the same ingredients, nutrition information, and serving size but had undergone multiple packaging changes with different release dates, the most recent product was included in the dataset whilst the older product was removed. For products that were classified under both demographic subcategories (e.g., ‘0–4 years’ and ‘5–12 years’), the product was removed from the ‘5–12 years’ subcategory to avoid duplication and underreporting in the ‘0–4 years’ subcategory. Products categorised as ‘maternal’ and ‘female’ in addition to being for children aged ‘0–4 years’ were removed (e.g., herbal tea targeted for consumption by a parent-to-be or parent). Growing Up Milks (GUMs), regulated under Standard 2.9.3 [36] as formulated supplementary foods for children aged 1–3 years and meant to provide a supplement to diets that may not be adequate to meet individual nutritional requirements, were included. Unlike infant formulas, GUMs have limited nutritional prescription (requiring only specified amounts of protein and energy, and only 1 vitamin or mineral) and have experienced significant growth in terms of their sales and marketing over the past decade [37], and there are no restrictions on how they are marketed. GUMs are not recommended by health authorities, and they are large contributors to free sugar intake in young children [38].

Drinkable products that were classified under ‘Food’ subcategories by Mintel were reclassified as ‘Drink’ to provide a more accurate representation of food products given that beverages are generally low in ED and may distort the overall ED means/medians [15,39,40]. For example, Baby Food products such as ‘Growing Up Milks’ and Dairy products such as ‘White Milk’ were reclassified as ‘Drink’ products. Variety packs were disaggregated for each flavour variety (e.g., honey, chocolate, and choc-chip) as nutrition data differed despite the food being the same, or similar, and packaged together. ED for each flavour was assessed, and the classification of ED and NOVA was determined.

The ED of a product was calculated using energy (kJ) and serving size (g or mL) located on nutrition information panels (NIPs) found on each product. Products with missing NIPs were rechecked on Mintel GNPD and the food/beverage company’s website. When NIPs were located, the ED was manually calculated. When NIPs were not available, the ED could not be manually calculated, and the item was removed from the dataset. On-pack ingredient lists were used to determine the NOVA classification of each item; all items presented product ingredient lists.

Note: All product classifications were agreed upon by the authors, KL, SD, and SM.

### 2.3. Nutrition Classifications

#### 2.3.1. ED Classification

The ED (kJ/100 g or kJ/100 mL) of each item was converted to kJ/g or kJ/mL and classified as low (≦4.184 kJ/g), medium (>4.184 kJ/g and <12.552 kJ/g), or high (≧12.552 kJ/g) based on the ED parameters specified by Raynor et al. [41].

#### 2.3.2. NOVA Classification

Products were classified into one of four NOVA groups specified by Monteiro et al. [31]: unprocessed or minimally processed (G1), processed culinary ingredients (G2), processed foods (G3), and ultra-processed foods (G4) [27]. Products were classified based on the presence of markers of ultra-processing (MUP) located in the ingredient list, as specified by Dickie et al. [42], i.e., cosmetic additives or processed food substances (Appendix A). When the ingredient list did not provide sufficient detail to reliably classify a food product, a conservative approach was taken wherein the product was classified into the lower processing level (i.e., G3 instead of G4) [42]. For example, potato starch was classified as G2 (culinary ingredient) unless specifically listed as ‘modified potato starch’, which is an MUP (G4). By doing this, the subjective overestimation of UPFs was reduced.

A random sample of products (10% of total dataset) was cross-checked for the accuracy of ED values, ED classification, and NOVA classification by two authors (KL and SD).

### 2.4. Statistical Analysis

All statistical analyses were conducted using STATA version 17 [43]. Descriptive analysis was performed for ED (frequency, median, and interquartile range (IQR)) and NOVA (frequency) for each ‘Food’ and ‘Drink’ subcategory. As the data were not normally distributed, non-parametric tests were employed. Pearson chi-squared tests were performed to assess whether ED differed between foods with lower levels of processing (G1/G2/G3 combined) and foods with a higher level of processing (G4). These tests were conducted separately for each age demographic.

## 3. Results

### 3.1. Category Distribution

A total of 1770 products were released between 1 May 2013 and 31 May 2023 (Figure 1) and distributed across 15 ‘Food’ (n = 1485, 84%) and 5 ‘Drink’ (n = 285, 16%) Mintel subcategories. Note: A total of seven ‘Drink’ subcategories were listed after drinkable ‘Food’ products were reclassified to ‘Drink’. For the total sample, 656 products were targeted to children ‘0–4 years’ (37%) and 1114 to children ‘5–12 years’ (63%). The majority of ‘Food’ (64%) and ‘Drink’ (60%) products were for children ‘5–12 years’ (Table 1).

For the total sample, the greatest proportion of ‘Food’ products were from Baby Food (30%), Snacks (14%), Sugar and Gum Confectionary (11%), and Bakery (9%). There were no products released under the Mintel subcategories Soup, Sweeteners and Sugar, and Sports and Energy Drinks. Baby Food contributed to 95% of all products for children aged ‘0–4 years’, with Growing Up Milks (n = 94) making up 15% of the total share of Baby Food (as ‘Drink’).

For children aged ‘5–12 years’, Snacks and Sugar and Gum Confectionary contributed the greatest to ‘Food’, with a 25% and 21% share, respectively. Other major contributors were Bakery (18%), Chocolate Confectionary (10%), Desserts and Ice Cream (9%), and Breakfast Cereals (8%), which collectively represented 91% of ‘Food’ products for this age group. Both Baby Food and Dairy contributed 35% and 31%, respectively, to the total sample of ‘Drink’ products, with 60% of all ‘Drink’ products targeted to children aged ‘5–12 years’ (Table 1).

### 3.2. Energy Density (kJ/g or kJ/mL)

The EDs for the total sample of products ranged from 0 to 25.56 kJ/g or mL. The median EDs for each ‘Food’ and ‘Drink’ subcategory are listed in Appendix A. Bakery (18.62 kJ/g), Breakfast Cereals (15.71 kJ/g), and Snacks (14.50 kJ/g) had the highest median EDs for children aged ‘0–4 years’. For children aged ‘5–12 years’, the subcategories with the highest median EDs were Hot Beverages (22.60 kJ/mL), Chocolate Confectionary (22.41 kJ/g), Bakery (18.34 kJ/g), Snacks (16.30 kJ/g), and Breakfast Cereals (16.10 kJ/g) (Table 1).

Sauces and Seasonings (3.14 kJ/g), Side Dishes (3.20 kJ/g), and Baby Food (3.23 kJ/g) had the lowest median EDs of the ‘Food’ subcategories in the ‘0–4 years’ age group. Baby Food (including Growing Up Milks) (2.82 kJ/mL) and Dairy (3.82 kJ/mL) had the lowest median EDs of the ‘Drink’ subcategories in the ‘0–4 years’ age group. The lowest median EDs for children aged ‘5–12 years’ were for Fruit and Vegetables (1.88 kJ/g) and Water (0.01 kJ/mL) (Table 1).

### 3.3. Energy Density Classification: Low, Medium, High

The proportions of low-, medium- and high-ED classifications for the total sample were 35%, 9%, and 56%, respectively (Appendix A). For children ‘0–4 years’, the majority (67%) of products were classified as low-ED, and for children ‘5–12 years’, the majority of products were classified as high-ED (65%). For children aged ‘0–4 years’, where the majority of products were considered Baby Food, 69% were classified as low-ED (Table 1).

The ‘Food’ subcategories in the ‘5–12 years’ age group that had the highest median EDs contained at least 85% of the products that were classified as high-ED. These subcategories were Chocolate Confectionary (n = 97/97 (100%); median ED = 22.41 kJ/g), Bakery (n = 143/166 (86.1%); median ED = 18.34 kJ/g), Snacks (n = 203/236 (86.0%); median ED = 16.30 kJ/g) and Sugar and Gum Confectionary (n = 184/195 (95.4%); median ED = 14.50 kJ/g) (Table 1).

### 3.4. NOVA Classification

The majority of products were classified as G4 for children aged ‘0–4 years’ (59%) and ‘5–12 years’ (93%) (Table 2). A greater proportion of products were classified as G1 in the ‘0–4 years’ age group (41%) compared to the ‘5–12 years’ age group (7%). Across the sample, the majority of products were classified as G4 in most subcategories, with the highest frequency under Baby Food (for both the ‘Food’ and ‘Drink’ subcategories), Bakery, Snacks, and Sugar and Gum Confectionary, with Breakfast Cereals, Chocolate Confectionary, Dairy (in both the ‘Food’ and ‘Drink’ subcategories), and Desserts and Ice Cream also contributing. The majority of Baby Food products under ‘Drink’ (i.e., Growing Up Milks) were classified as G4 (93%).

For children aged ‘5–12 years’, the Bakery, Snacks, and Sugar and Gum Confectionary subcategories reported the highest frequency of G4 products, with Breakfast Cereals, Chocolate Confectionary, and Desserts and Ice Cream also notably contributing (a cumulative 92% of G4 products for children aged ‘5–12 years’). For the ‘Drink’ subcategories across the ‘Total Sample’, 89% were classified as G4, with Baby Food and Dairy contributing 37% and 33% of G4 products, respectively (Table 2).

### 3.5. NOVA Classification and Energy Density

For each ED classification (low, medium, high), the majority of products in both the ‘0–4 years’ and ‘5–12 years’ groups were classified as G4, except in the ‘0–4 years’ group, where 40% of low-ED products were classified as G4. For all products in the ‘5–12 years’ group, 93% were classified as G4, with 62% classified as G4 and high-ED (Table 3). Examples of ‘Food’ and ‘Drink’ subcategories according to the ED and NOVA classifications are listed in Table 4.

Pearson’s chi-squared tests revealed statistically significant differences in the distribution of ED categorisation between the NOVA groups for ‘Food’ in the ‘0–4 years’ age group (χ^2^(2) = 26.5620, *p* < 0.001). Approximately 22% (57 of 262) of G1/G2/G3 foods were high-ED, while 42% (119 of 280) of G4 foods were high-ED. Statistically significant differences in the distribution of ED between NOVA groups were not observed for ‘Food’ for the ‘5–12 years’ group (χ^2^(2) = 0.1896, *p* = 0.663). The majority of G1/G2/G3 (79%; 679 of 891) ‘Food’ products for the ‘5–12 years’ group were classified as high-ED. Statistically significant differences in the distribution of ED between NOVA groups were not observed for ‘Drinks’ for the ‘5–12 years’ group (χ^2^(2) = 1.3702, *p* = 0.242), with 100% of G1/G2/G3 ‘Drinks’ and 95% of G4 ‘Drinks’ classified as low- or medium-ED. The assumptions for Pearson’s chi-squared testing were not met for ‘Drinks’ in the ‘0–4 years’ age group due to the small sample size. For the ‘0–4 years’ age group, foods categorised as G1/G2/G3 had a combined median ED of 2.8 kJ/g, while those categorised as G4 had a combined median ED of 4.3 kJ/g. For the ‘5–12 years’ age group, foods categorised as G1/G2/G3 had a combined median ED of 16.1 kJ/g, while those categorised as G4 had a combined median ED of 15.5 kJ/g.

## 4. Discussion

This study is the first to explore the energy density and level of processing of packaged food and beverage products sold for consumption by Australian children, aged 0–12 years. Over half of the products released into the market in the previous 10 years were classified as high-ED, with most of those products intended for children aged ‘5–12 years’. An overwhelming proportion of products were classified as UPFs across ED classifications (low, medium, or high). There was evidence of statistically significant differences in the ED classification between NOVA groups for ‘Food’ in the ‘0–4 years’ group, but not for children aged ‘5–12 years’. There are limited studies with which to compare our results. Only one previous study has analysed ED in products for Australian children using the Mintel GNPD. Azzopardi et al. [44] reported that 74% of foods (excluding beverages) released between 2014 and 2018 and targeted to children (5–12 years) in Australia were classified as high-ED [44]. This current study builds on knowledge by expanding the dataset to incorporate both ‘Food’ and ‘Drink’ products over a 10-year timeframe and two different age demographics. The slightly lower proportion of high-ED ‘Food’ classifications in our study can likely be explained by different ED parameters being used (e.g., ‘Food’ with ED > 9.5 kJ/g was classified as high-ED compared to food with ED ≧ 12.552 kJ/g in the present study) [44]. Crino et al. [45] analysed the healthiness of packaged foods and drinks in Australia using the George Institute’s FoodSwitch database. For the food subcategories Snacks, Cereal and Grain Products, and Bread and Bakery Products, the median EDs were all above the high-ED cut-off (applying the ED parameters used in our study). The median EDs for the abovementioned food subcategories closely align with the results obtained in our study. Furthermore, Sutton et al. [46] analysed the US food system over a 30-year period and reported that 47% of foods were classified as high-ED (mostly due to their high sugar content), which closely aligns with our study. Whilst Sutton et al. [46] analysed all foods (including unpackaged foods), the parameters used to classify high-ED foods (>8.368 kJ/g) were lower than the specifications used in our study.

A large majority of the products examined in the present study were classified as UPFs, which aligns with other studies analysing packaged food products in the Australian marketplace [45,47]. For children aged ‘0–4 years’, 95% of products were categorised as Baby Food, with 57% of those products classified as UPFs (including Growing Up Milks but excluding Baby Formulas). This statistic is concerning. Infancy is an important period where a child’s diet transitions from human milk or infant formula to a complimentary feeding regime [6,18]. Infants and young children are the lowest consumers of UPFs, which suggests that they consume mostly non-UPFs, particularly in their early years of life [48]. However, global sales trends report an increased consumption of processed baby foods [49]. The consumption of UPFs increases exposure to industrial additives (i.e., sensory-related industrial additives (SRIAs)), which are used to extend shelf-life but also to enhance the sensory qualities of foods, such as texture, flavour, sweetness, and appearance [29]. Early exposure to these highly palatable foods may disrupt appetite regulation [32] and could lead to overconsumption and weight accumulation in children [28,33], with childhood obesity being highly predictive of adolescent and adulthood obesity [50]. A study conducted in adults demonstrated the effects of UPFs on weight gain by highlighting a possible weakened satiety response leading to overeating, despite the diets being matched for energy, palatability, and macronutrient composition [13]. This raises serious concerns regarding early exposure to UPFs among young children.

The vast majority of products intended for children aged ‘5–12 years’ were UPFs, with a notable increase in the percentage of UPF products for this age group compared with the younger age group. The Australian population (2+ years) consumes 42% of their total daily EI through diet from UPFs [11], which is not surprising given the prevalence and level of processing of packaged products on the Australian marketplace. Population-based cross-sectional studies demonstrate that the increasing dietary share of UPFs (often) displaces nutrient-rich, non-UPF consumption [11,27,40,51]. The availability of UPF products in the food supply intended for consumption by older children (compared to younger children), as seen in our results, thus could contribute to the displacement of nutritious minimally processed foods in children’s diets. Other studies have also observed a clear trend of increasing UPF consumption with increasing age from childhood to adolescence [40], with school-aged children and adolescents being the highest consumers of UPFs in Australia [40,48]. An opportunity exists to emphasise the importance of limiting UPF exposure and consumption among young children and promoting the provision of nutrient-rich non-UPFs from home-cooked meals and snacks to improve nutrient intake, manage energy balance, and improve overall diet quality [52].

The differences in ED classification between NOVA groups presented mixed results in the present study. Statistical significance was reported for ‘Food’ products for children aged ‘0–4 years’, which is supported by other studies [45,47]. This was likely due to large proportions of low-ED, non-UPF (e.g., pureed fruit and vegetable pouches, plain yoghurt) and high-ED UPFs (e.g., sweetened biscuits, oat bars). For ‘Food’ products intended for children aged ‘5–12 years’, differences in ED classification between NOVA groups were not observed, which was unexpected. This is because the majority of products across all ED classifications were UPFs (66% of low-ED, 94% of medium-ED, 89% of high-ED foods), with fewer products falling into the other categories. This also highlights that those foods classified as low-ED and UPF, which are typically deemed ‘healthy’ by the ADGs (such as dairy, fruit, and vegetable snacks) based on nutrient and energy recommendations [6], may have a degraded food matrix and/or contain SRIAs, which may potentially result in overconsumption of UPFs. Regularly consumed Australian products such as breakfast cereals, flavoured yoghurts, and mass-produced breads are often ultra-processed (and highly palatable) [11], which is misleading for consumers as the presence of SRIAs likely lessens the nutritional quality of the food.

This study is the first to concurrently examine the ED and level of processing of food and beverage products intended for children and released into the Australian marketplace over the past 10 years. A major strength of this study was using the Mintel GNPD, which provides current information on newly released products and provides a snapshot of product activity over a specified timeframe. Furthermore, the differentiation of products by age demographics enabled direct comparison and insight into the products intended for children based on age-related positioning claims. There are limitations to this study that warrant mentioning. The Mintel GNPD likely represents a large proportion of the current Australian marketplace, with the inclusion of products from all the major supermarkets; however, as only newly released products were added, some products on the marketplace directed to children could have been missed. Additionally, categorising products through the NOVA classification raises some criticism based on the subjectivity of ‘processing’ definitions, which was reduced by using a more objective classification process that identified MUPs listed in the product ingredient list to ensure the robustness of the data.

There are several implications of this research. Government action is needed to reduce the prevalence and consumption of energy-dense and ultra-processed products on the Australian food retail marketplace. Population-level initiatives that not only target the reduction in consumption of high-ED products and UPFs (e.g., regulation of health claims and marketing regarding packaging [53] and laws that prohibit the promotion of infant formulas and Growing Up Milks [54], etc.) but also prioritise the promotion of consuming minimally processed and nutritious foods (e.g., healthy food procurement policies in public institutions) are equally important for improving public health obesity outcomes in children. Extending this research with qualitative research targeting parents’ food choices and their understanding of both ED and UPF classifications will provide further insight into the drivers of product selection and consumer knowledge.

## 5. Conclusions

The majority of newly released food products on the Australian marketplace intended for consumption by children aged ‘5–12 years’ are classified as high-ED and ultra-processed. Reducing exposure to and consumption of these products are important in managing EI, improving diet quality, and decreasing the risk of developing overweight and obesity in children. This study provides evidence to inform policy that aims to improve the healthiness of the food environment for children in Australia and to tighten regulations on packaged foods and beverages targeted to children.

## Figures and Tables

**Figure 1 nutrients-17-02293-f001:**
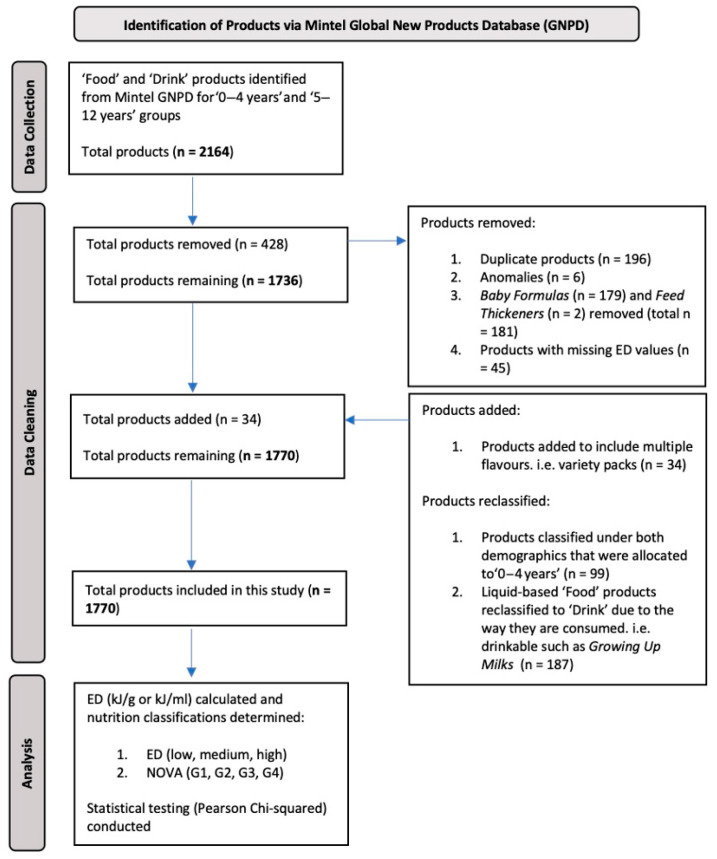
Flowchart demonstrating the process of data collection, cleaning, and determination of ED and NOVA classifications of all ‘Food’ and ‘Drink’ products targeted to ‘0–4 years’ and ‘5–12 years’ demographics and reported by the Mintel Global New Products Database as being released in Australia (2013–2023).

**Table 1 nutrients-17-02293-t001:** Descriptive data for each ‘Food’ and ‘Drink’ subcategory by group (‘0–4 years’ and ‘5–12 years’), reported by the Mintel Global New Products Database as being released in Australia (2013–2023).

	‘0–4 Years’								‘5–12 Years’							
	n	ED (kJ/g or mL)							n	ED (kJ/g or mL)						
Food	Total (n)	Low ED	Med ED	High ED	Median	IQR	Min	Max	Total (n)	Low ED	Med ED	High ED	Median	IQR	Min	Max
Baby Food *	520	329	30	161	3.23	12.38	1.28	23.80	0	0	0	0	0	0	0	0
Bakery	1	0	0	1	18.62	0	18.62	18.62	166	0	23	143	18.34	4.27	8.07	22.46
Breakfast Cereals	3	0	0	3	15.71	0.36	15.39	15.76	80	1	1	78	16.10	0.65	4.11	19.70
Chocolate Confectionary	0	0	0	0	0	0	0	0	97	0	0	97	22.41	1.52	16.27	24.87
Dairy *	1	0	0	1	12.80	0	12.80	12.80	27	12	6	9	5.96	11.08	3.40	17.90
Desserts and Ice Cream	0	0	0	0	0	0	0	0	91	40	50	1	4.64	3.31	0.06	13.00
Fruit and Vegetables	0	0	0	0	0	0	0	0	7	7	0	0	1.88	1.98	1.38	4.06
Meals and Meal Centres	3	0	3	0	6.45	1.77	4.68	6.45	14	11	3	0	3.04	0.47	2.32	9.56
Processed Fish, Meat and Pork Products	0	0	0	0	0	0	0	0	19	0	19	0	7.46	1.61	4.24	9.83
Sauces and Seasonings	2	2	0	0	3.14	0.13	3.07	3.20	2	2	0	0	3.53	1.05	3.00	4.05
Savoury Spreads	0	0	0	0	0	0	0	0	5	0	3	2	11.15	5.57	6.40	17.05
Side Dishes	1	1	0	0	3.20	0	3.20	3.20	1	0	1	0	7.95	0	7.95	7.95
Snacks	11	0	1	10	14.50	3.45	8.64	19.27	236	22	11	203	16.30	4.55	2.33	25.56
Sugar and Gum Confectionary	0	0	0	0	0	0	0	0	195	2	9	184	14.50	1.46	1.43	20.07
Sweet Spreads	0	0	0	0	0	0	0	0	3	0	0	3	14.20	8.00	14.13	22.13
‘Food’ Total	542	332	34	176	3.32	12.47	1.28	23.80	943	97	126	720	15.5	5.39	0	25.56
Drink	Total (n)	Low ED	Med ED	High ED	Median	IQR	Min	Max	Total (n)	Low ED	Med ED	High ED	Median	IQR	Min	Max
Baby Food *	100	100	0	0	2.82	0.27	0	28.00	0	0	0	0	0	0	0	0
Carbonated Soft Drinks	0	0	0	0	0	0	0	0	7	7	0	0	1.94	1.30	0.75	2.24
Dairy *	9	8	1	0	3.82	0.53	3.49	4.21	78	78	0	0	2.90	1.06	1.84	3.99
Hot Beverages	0	0	0	0	0	0	0	0	5	1	0	4	22.60	0	2.00	22.60
Juice Drinks	0	0	0	0	0	0	0	0	46	46	0	0	1.23	0.62	0.59	2.86
Nutritional Drinks and Other Beverages	5	2	2	1	4.19	0.01	3.90	17.00	29	22	3	4	2.80	1.73	0.12	19.13
Water	0	0	0	0	0	0	0	0	6	6	0	0	0.01	0.32	0	0.33
‘Drink’ Total	114	110	3	1	2.86	0.31	0	28.00	171	160	3	8	2.46	1.49	0	22.60
Total	656	442	37	177	3.08	11.45	0	28.00	1114	250	129	728	14.52	12.56	0	0

* This subcategory includes some liquid products that the Mintel Global New Products Database considered ‘Food’ that were reclassified as ‘Drink’ due to the way they are consumed, e.g., in the ‘Food’ subcategory, Baby Food products such as Growing Up Milks were reclassified under ‘Drink’, and Baby Food and Drinkable Yoghurt and Liquid Cultures, Flavoured Milk, Plant-Based Drink, and White Milk in the ‘Food’ Dairy subcategory were reclassified under the ‘Drink’ Dairy subcategory. n: number of items; ED: energy density; IQR: interquartile range. ED classified as low (≦4.184 kJ/g), medium (>4.184 kJ/g and <12.552 kJ/g), or high (≧12.552 kJ/g).

**Table 2 nutrients-17-02293-t002:** Frequency (n) of each ‘Food’ and Drink’ subcategory by group (‘0–4 years’, ‘5–12 years’ and ‘Total Sample’) and NOVA classification reported by the Mintel Global New Products Database as being released in Australia (2013–2023).

NOVA Classification						
	‘0–4 Years’		‘5–12 Years’		‘Total Sample’	
	G1/G2/G3 Combined	G4	G1/G2/G3 Combined	G4	G1/G2/G3 Combined	G4
Food	n	n	n	n	n	n
Baby Food *	257	263	0	0	257	263
Bakery	0	1	4	162	4	163
Breakfast Cereals	0	3	2	78	2	81
Chocolate Confectionary	0	0	0	97	0	97
Dairy *	1	0	6	21	7	21
Desserts and Ice Cream	0	0	0	91	0	91
Fruit and Vegetables	0	0	2	5	2	5
Meals and Meal Centres	2	1	0	14	2	15
Processed Fish, Meat and Pork Products	0	0	0	19	0	19
Sauces and Seasonings	0	2	0	2	0	4
Savoury Spreads	0	0	0	5	0	5
Side Dishes	0	1	0	1	0	2
Snacks	2	9	37	199	39	208
Sugar and Gum Confectionary	0	0	0	195	0	195
Sweet Spreads	0	0	1	2	1	2
‘Food’ Total	262	280	52	891	314	1171
Drink	n	n	n	n	n	n
Baby Food *	7	93	0	0	7	93
Carbonated Soft Drinks	0	0	0	7	0	7
Dairy *	0	9	3	75	3	84
Hot Beverages	0	0	0	5	0	5
Juice Drinks	0	0	18	28	18	28
Nutritional Drinks and Other Beverages	0	5	0	29	0	34
Water	0	0	3	3	3	3
‘Drink’ Total	7	107	24	147	31	254
Total	269 (41%)	387 (59%)	76 (7%)	1038 (93%)	345 (19%)	1425 (81%)

* This subcategory includes some liquid products that the Mintel Global New Products Database considered ‘Food’ that were reclassified as ‘Drink’ due to the way they are consumed, e.g., in the ‘Food’ subcategory, Baby Food products such as Growing Up Milks were reclassified under ‘Drink’ and Baby Food and Drinkable Yoghurt and Liquid Cultures, Flavoured Milk, Plant-Based Drinks, and White Milk under the ‘Food’ Dairy subcategory were reclassified under the ‘Drink’ Dairy subcategory. Percentage values are rounded up to whole numbers. n: number of items; G1: unprocessed or minimally processed; G2: processed culinary ingredient; G3: processed; G4: ultra-processed.

**Table 3 nutrients-17-02293-t003:** Frequency (n) and median of EDs by ED classification (low, medium, high) and NOVA classification for each group (‘0–4 years’ and ‘5–12 years’) and ‘Food’ and ‘Drink’ category reported by the Mintel Global New Products Database as being released in Australia (2013–2023).

	‘0–4 Years’												‘5–12 Years’											
	‘Food’						‘Drink’						‘Food’						‘Drink’					
	n			Median ED (kJ/g or mL)			n			Median ED (kJ/g or mL)			n			Median ED (kJ/g or mL)			n			Median ED (kJ/g or mL)		
	Low ED	Med ED	High ED	Low ED	Med ED	High ED	Low ED	Med ED	High ED	Low ED	Med ED	High ED	Low ED	Med ED	High ED	Low ED	Med ED	High ED	Low ED	Med ED	High ED	Low ED	Med ED	High ED
G1, G2, and G3 combined	198	7	57	2.63	4.68	17.00	7	0	0	2.88	0	0	8	3	41	2.56	12.35	16.83	24	0	0	1.44	0	0
G4	134	27	119	3.05	4.70	16.53	103	3	1	2.83	4.19	17.00	89	123	679	3.10	7.95	16.67	136	3	8	2.51	4.63	20.87

n: number of items; ED: energy density; G1: unprocessed or minimally processed; G2: processed culinary ingredient; G3: processed; G4: ultra-processed.

**Table 4 nutrients-17-02293-t004:** List of ‘Food’ and Drink’ subcategories by group (‘0–4 years’ and ‘5–12 years’), ED classification, and NOVA classification reported by the Mintel Global New Products Database as being released in Australia (2013–2023).

		‘0–4 Years’	‘5–12 Years’
ED Classification	NOVA Classification	Mintel GNPD Food and Drink Subcategories	
Low	G1/G2/G3 combined	Baby Cereals Baby Fruit Products, Desserts and Yoghurts Baby Savoury Meals and Dishes Growing Up Milks	Dairy Fruit and Vegetables Juice Drinks Snacks Water
Low	G4	Baby Cereals Baby Fruit Products, Desserts and Yoghurts Baby Savoury Meals and Dishes Growing Up Milks	Carbonated Soft Drinks Drinking Yoghurt and Liquid Cultured Milk Flavoured Milk Spoonable Yoghurt Dessert Subcategories Nectars Instant Noodles Prepared Meals Beverage Mixes Fruit Snacks
Medium	G1/G2/G3 combined	Baby Fruit Products, Desserts and Yoghurts Prepared Meals	Soft Cheese and Semi-Soft Cheese
Medium	G4	Baby Fruit Products, Desserts and Yoghurts Prepared Meals Baby Savoury Meals and Desserts Baby Snacks	Bread and Bread Products Chilled Desserts Dairy Based Ice Cream and Frozen Yoghurt Meat and Poultry Products Sticks, Liquids and Sprays
High	G1/G2/G3 combined	Baby Biscuits and Rusks Baby Fruit Products, Desserts and Yoghurts Baby Snacks	Baking Ingredients and Mixes Hard Cheese and Semi Hard Cheese Fruit Snacks Popcorn Rice Snacks Snack/Cereal/Energy Bars Vegetable Snacks
High	G4	Baby Biscuits and Rusks Baby Snacks Snack/Cereal/Energy Bars	Baking Ingredients and Mixes Bread and Bread Products Cakes, Pastries and Sweet Goods Sweet Biscuits/Cookies Cold Cereals Chocolate Subcategories Processed Cheeses Malt and Other Hot Beverages Fruit Snacks Popcorn Potato Snacks Rice Snacks Snack/Cereal/Energy Bars Wheat and Other Grain-Based Snacks Sugar and Gum Confectionary Subcategories

The Mintel GNPD Food and Drink subcategories listed do not make up the complete list but form the majority of product subcategories in each NOVA and ED classification. G1: unprocessed or minimally processed; G2: processed culinary ingredient; G3: processed; G4: ultra-processed.

## Data Availability

The data that support the findings of this study are available from Mintel Group Ltd. but the restrictions apply to the availability of these data, which were used under license for the current study, and so are not publicly available.

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
