# Peer review of "Energy Density and Level of Processing of Packaged Food and Beverages Intended for Consumption by Australian Children"

_nutrients, 2025, doi:10.3390/nu17142293_

Round 1
Reviewer 1 Report
Comments and Suggestions for Authors
Title: “Energy Density and Level of Processing of Packaged Food and Beverages Intended for Consumption by Australian Children”
# Global evaluation:
The present manuscript proffers a highly relevant and timely investigation into the issue of childhood obesity by examining dietary exposures, with a particular focus on packaged food products intended for children. A notable strength of the study lies in the utilisation of the Mintel Global New Products Database, which encompasses a substantial 10-year period. This extensive dataset provides valuable breadth and depth, allowing the authors to draw meaningful insights into long-term trends.
A key methodological strength lies in the dual analysis of energy density (ED) and food processing level using the NOVA classification. This combined approach offers a novel and rigorous framework for evaluating the nutritional quality of food and beverage products. The manuscript is further supported by the presence of well-structured tables and figures, which present the data in a clear and easily interpretable manner for the reader.
Nevertheless, the manuscript also exhibits certain deficiencies that must be addressed. While the study is rich in descriptive detail, it lacks a strong analytical or policy modelling component that would deepen the interpretation of the results and enhance its practical impact. Moreover, the utilisation of the term "high-ED" as an inherently negative marker oversimplifies the assessment of nutritional quality, as certain high-energy-dense foods, including nuts and specific dairy products, have the capacity to promote health. The discussion occasionally adopts an advocacy stance, making assertive claims that are not consistently supported by evidence or counterpoints. Finally, the manuscript features repeated definitions and methodological descriptions, contributing to redundancy and reducing overall conciseness. The resolution of these issues would result in a substantial enhancement of the clarity and impact of the study.
# Suggestions by lines:
Line 14. “Higher energy density (ED; kJ/g) and higher levels of processing...” Please, add “have been associated” to avoid implying causality.
Line 19. Formatting error: replace “<12.552kJ/g” with “≥4.184kJ/g and <12.552kJ/g”.
Line 30. "Greater consideration of ‘healthy’ low-ED UPFs..." needs to be clarified: are you implying some UPFs can be "healthy"? Contradicts the tone of rest of paper.
Lines 33–35. “One quarter of children are living with overweight or obesity...” – the data reference is from 2011 (Ref 2). Please, include more recent data.
Line 37. Misuse of parentheses; the list inside should be separated: "i.e., foods high in sugar, saturated fats, or energy density..."
Line 74–83. Good rationale for the study, but would benefit from stating explicitly what gap in current literature this fills. Is it the first study post-ADG?
Line 96. Suggestion. Change “were analysed” to “were obtained and analysed” for clarity.
Line 114. “...based on packaging attributes...” . Please clarify whether this includes only front-of-pack claims or ingredient lists too.
Lines 139–140. Exclusion of baby formula is justified, but it contradicts the inclusion of Growing Up Milks (GUMs), which are also tightly regulated and controversial in marketing practices. This needs discussion.
Line 157. “...the item was removed from the dataset”. Please report how many such products were excluded due to this.
Lines 188–199. Figure 1 should be referenced in-text explicitly. Also, the flowchart is essential and should be included early in Results.
Line 235–237. All percentages should be reported with decimal precision or consistently rounded. For instance, “n=184/195” should include a percentage (94.4%).
Table 1 is comprehensive but difficult to digest; consider summarizing key trends per group before diving into specifics.
Line 287. “...intended for consumption by Australian children...” . There is a slightly awkward repetition from title; could say “sold for consumption by...”
Lines 320–337. Excellent integration of findings with developmental considerations (appetite regulation in early childhood).
Line 355–370. Important point about “healthy”-perceived UPFs — this section could be strengthened with concrete examples and dietary guideline contrasts.
# Conceptual issues and recommendations
-On the Role of UPFs
There’s an implicit assumption that all UPFs are bad. Some nuance would be helpful. Are there UPFs that fulfill specific nutritional roles (e.g., fortification)?
Explore the role of reformulation policies (e.g., Health Star Rating in Australia) as a moderating factor.
-Energy Density and Nutrient Density
The text sometimes treats ED as a negative trait. But low-ED foods may be nutrient-poor (e.g., sugar-free jelly), and high-ED foods may be nutrient-rich (e.g., avocado).
It is suggested to add a short paragraph clarifying this complexity and how the policy should respond.
-Policy Relevance
Recommendations are too general. Propose specific policies: e.g., front-of-pack ED labelling, bans on cartoon marketing for G4 products, or aligning marketing permissions with NOVA classification.
# Writing recommendations
-Avoid repetition, ED and NOVA definitions appear in the abstract, intro, and methods. Consolidate to one clear definition each.
-Clarify statistical methods, describe assumptions of the chi-square test, especially when assumptions weren’t met (as noted in line 279).
-Be cautious about using terms like “urgent” or “critical” without data quantifying harm. This is especially important for a nutrition journal audience that values neutrality and data-driven language.
-References. Ensure all references are recent. A few (e.g., ref #2 from 2011) are outdated for the current prevalence.
# Suggested improvements
-Add a flow diagram in Methods showing total products retrieved, removed (e.g., duplicates, no NIP), and final sample.
-Refine Tables. Consider simplifying Table 1 into age-specific summaries and moving extensive ED and NOVA breakdowns to supplementary materials.
-Broaden the framing. Include a brief discussion on how Australia compares to other countries (e.g., UK, US) in child-targeted food ED and UPF trends.
-Add sensitivity analysis: What happens if “borderline” NOVA classifications were assigned to higher/lower categories?
-Clarify ambiguous terms. For instance, “Growing Up Milks”: include a footnote defining this product and its nutritional context.
Reviewer 2 Report
Comments and Suggestions for Authors
This is an interesting research article with adequate novelty. Some points should be addressed.
- In the Introduction section, the authors should add the exact prevelance of overweight and obesity separetely with numbers %.
- Childhood obesity is a risk factor for persisted obesity in the adolescence and adulthood which is associated with several chronic diseases, such as cardiobascular diseases, etc. The above issue should be added in the 1st paragraph of the Introduction.
- The authors should add a bit more information concerning the disruption of appetite regulation in line 71.
- In section 2.4, the normality distribution test and the software used should be reported.
- Tables 3 and 4 should be moved to the Results section.
- The authors should add at the end of the Discussion section a paragraph with the strengths and the limitations of our study.
Reviewer 3 Report
Comments and Suggestions for Authors
Interesting manuscript entitled:
Energy Density and Level of Processing of Packaged Food and 2 Beverages Intended for Consumption by Australian Children.
This study has several strengths:
This study addresses a critical and underexplored dimension of the children's food environment, especially in the Australian market, using key food quality indicators (ED and NOVA), it also has a robust Database, the use of Mintel GNPD provides a representative sample of new products launched in the country, allowing for an up-to-date view of the market.
However, I believe that the discussion should be improved, for example, according to high quality evidence, is it possible to indicate that infant foods classified as ultra-processed are a risk factor for obesity in children, for example breakfast cereals, yoghurt or flavoured milks, sugary drinks or juices or biscuits, when evaluated separately, i.e. evidence of yoghurt with obesity or breakfast cereals with childhood obesity? all these foods have the same risk or is it necessary to analyse them separately, even if they are classified as ultra-processed. Studies in adult populations show, for example, that for diabetes and cardiovascular disease risk, sugar-sweetened beverages and processed meats are consistent with increased risk, and even foods classified as ultra-processed are protective factors.
Is there any information on the % caloric intake of these groups of ultra-processed products in the intake of Australian infants and children? This is also an interesting point to discuss, as it is not the same an ultra-processed product rich in saturated fats or sugar that is consumed every day and in large quantities as others that might be consumed only once a month.
Round 2
Reviewer 1 Report
Comments and Suggestions for Authors
The manuscript has been significantly improved and is ready for publication in its current form.
Reviewer 2 Report
Comments and Suggestions for Authors
The authors have significantly revised and improved their manuscript.